# SPIGAN: Privileged Adversarial Learning from Simulation

**Kuan-Hui Lee, Jie Li, Adrien Gaidon**
Toyota Research Institute
{kuan.lee,jie.li,adrien.gaidon}@tri.global

**& German Ros**
Intel Labs
german.ros@intel.com

## Abstract

Deep Learning for Computer Vision depends mainly on the source of supervision. Photo-realistic simulators can generate large-scale automatically labeled synthetic data, but introduce a domain gap negatively impacting performance. We propose a new unsupervised domain adaptation algorithm, called SPIGAN, relying on Simulator Privileged Information (PI) and Generative Adversarial Networks (GAN). We use internal data from the simulator as PI during the training of a target task network. We experimentally evaluate our approach on semantic segmentation. We train the networks on real-world Cityscapes and Vistas datasets, using only unlabeled real-world images and synthetic labeled data with z-buffer (depth) PI from the SYNTHIA dataset. Our method improves over no adaptation and state-of-the-art unsupervised domain adaptation techniques.

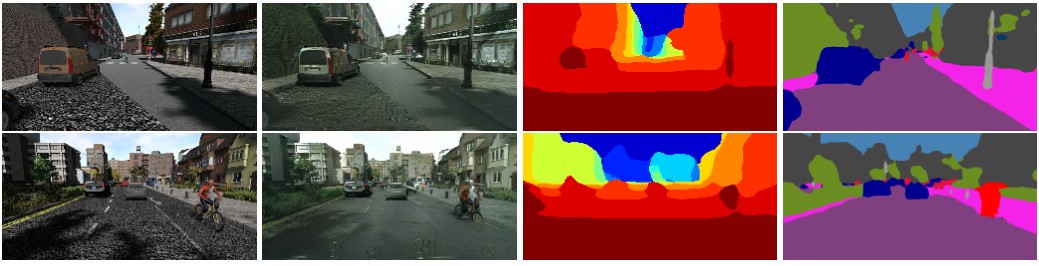

Figure 1: SPIGAN example inputs and outputs. From left to right: input images from a simulator; adapted images from SPIGAN's generator network; predictions from SPIGAN's privileged network (depth layers); semantic segmentation predictions from the target task network.

## 1 Introduction

Learning from as little human supervision as possible is a major challenge in Machine Learning. In Computer Vision, labeling images and videos is the main bottleneck towards achieving large scale learning and generalization. Recently, training in simulation has shown continuous improvements in several tasks, such as optical flow (Mayer et al., 2016), object detection (Marín et al., 2010; Vazquez et al., 2014; Xu et al., 2014; Sun & Saenko, 2014; Peng et al., 2015), tracking (Gaidon et al., 2016), pose and viewpoint estimation (Shotton et al., 2011; Papon & Schoeler, 2015; Su et al., 2015), action recognition (de Souza et al., 2017), and semantic segmentation (Handa et al., 2016; Ros et al., 2016; Richter et al., 2016). However, large domain gaps between synthetic and real domains remain as the main handicap of this type of strategies. This is often addressed by manually labeling some amount of real-world target data to train the model on mixed synthetic and real-world labeled data (supervised domain adaptation). In contrast, several recent *unsupervised* domain adaptation algorithms have leveraged the potential of Generative Adversarial Networks (GANs) (Goodfellow et al., 2014) for pixel-level adaptation in this context (Bousmalis et al., 2017; Shrivastava et al., 2016). These methods often use simulators as black-box generators of $(x, y)$ input / output training samples for the desired task.

Our main observation is that simulators internally know a lot more about the world and how the scene is formed, which we call Privileged Information (PI). This Privileged Information includes

physical properties that might be useful for learning. This additional information $z$ is not available in the real-world and is, therefore, generally ignored during learning. In this paper, we propose a novel adversarial learning algorithm, called *SPIGAN*, to leverage Simulator *PI* for GAN-based *unsupervised* learning of a target task network from *unpaired unlabeled* real-world data.

We jointly learn four different networks: (i) a generator $G$ (to adapt the pixel-level distribution of synthetic images to be more like real ones), (ii) a discriminator $D$ (to distinguish adapted and real images), (iii) a task network $T$ (to predict the desired label $y$ from image $x$), and (iv) a *privileged network* $P$ trained on both synthetic images $x$ and adapted ones $G(x)$ to predict their associated privileged information $z$. Our main contribution is a new method to leverage *PI* from a simulator via the privileged network $P$, which acts as an auxiliary task and regularizer to the task network $T$, the main output of our SPIGAN learning algorithm.

We evaluate our approach on semantic segmentation in urban scenes, a challenging real-world task. We use the standard Cityscapes (Cordts et al., 2016) and Vistas (Neuhold et al., 2017) datasets as target real-world data (without using any of the training labels) and SYNTHIA (Ros et al., 2016) as simulator output. Although our method applies to any kind of PI that can be predicted via a deep network (optical flow, instance segmentation, object detection, material properties, forces, ...), we consider one of the most common and simple forms of PI available in any simulator: depth from its z-buffer. We show that SPIGAN can successfully learn a semantic segmentation network $T$ using no real-world labels, partially bridging the sim-to-real gap (see Figure 1). SPIGAN also outperforms related state-of-the-art unsupervised domain adaptation methods.

The rest of the paper is organized as follows. Section 2 presents a brief review of related works. Section 3 presents our SPIGAN unsupervised domain adaptation algorithm using simulator privileged information. We report our quantitative experiments on semantic segmentation in Section 4, and conclude in Section 5.

## 2 RELATED WORK

Domain adaptation (cf. Csurka (2017) for a recent review) is generally approached either as domain-invariant learning (Hoffman et al., 2013; Herath et al., 2017; Yan et al., 2017; Ganin & Lempitsky, 2015) or as a statistical alignment problem (Tzeng et al., 2014; Long et al., 2015). Our work focuses on *unsupervised* adaptation methods in the context of deep learning. This problem consists in learning a model for a task in a target domain (e.g., semantic segmentation of real-world urban scenes) by combining *unlabeled* data from this domain with labeled data from a related but different *source* domain (e.g., synthetic data from simulation). The main challenge is overcoming the domain gap, i.e. the differences between the source and target distributions, without any supervision from the target domain. The Domain Adversarial Neural Network (DANN) (Tzeng et al., 2014; Ganin & Lempitsky, 2015; Ganin et al., 2016) is a popular approach that learns domain invariant features by maximizing domain confusion. This approach has been successfully adopted and extended by many other researchers, e.g., Purushotham et al. (2017); Chen et al. (2017); Zhang et al. (2017). Curriculum Domain Adaptation (Zhang et al., 2017) is a recent evolution for semantic segmentation that reduces the domain gap via a curriculum learning approach (solving simple tasks first, such as global label distribution in the target domain).

Recently, adversarial domain adaptation based on GANs (Goodfellow et al., 2014) have shown encouraging results for unsupervised domain adaptation directly at the pixel level. These techniques learn a generative model for source-to-target image translation, including from and to multiple domains (Taigman et al., 2016; Shrivastava et al., 2016; Zhu et al., 2017; Isola et al., 2017; Kim et al., 2017). In particular, CycleGAN (Zhu et al., 2017) leverages cycle consistency using a forward GAN and a backward GAN to improve the training stability and performance of image-to-image translation. An alternative to GAN is Variational Auto-Encoders (VAEs), which have also been used for image translation (Liu et al., 2017).

Several related works propose GAN-based unsupervised domain adaptation methods to address the specific domain gap between synthetic and real-world images. SimGAN (Shrivastava et al., 2016) leverages simulation for the automatic generation of large annotated datasets with the goal of refining synthetic images to make them look more realistic. Sadat Saleh et al. (2018) effectively leverages synthetic data by treating foreground and background in different manners. Similar to our approach,

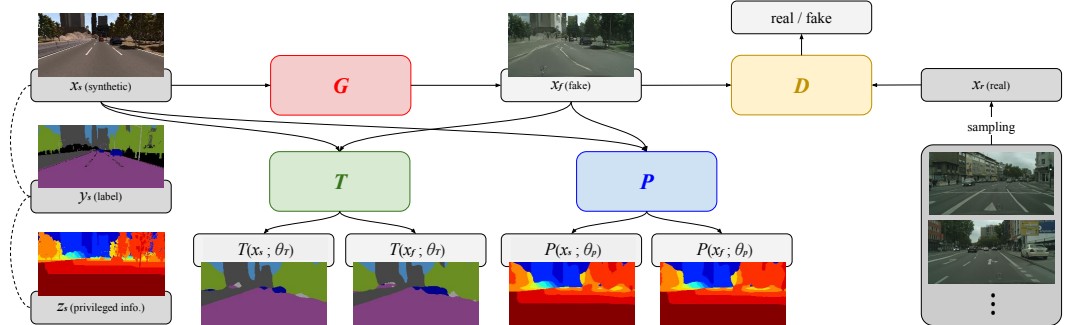

Figure 2: SPIGAN learning algorithm from unlabeled real-world images $x_r$ and the unpaired output of a simulator (synthetic images $x_s$, their labels $y_s$, e.g. semantic segmentation ground truth, and Privileged Information PI $z_s$, e.g., depth from the z-buffer) modeled as random variables. Four networks are learned jointly: (i) a generator $G(x_s) \sim x_r$, (ii) a discriminator $D$ between $G(x_s) = x_f$ and $x_r$, (iii) a perception task network $T(x_r) \sim y_r$, which is the main target output of SPIGAN (e.g., a semantic segmentation deep net), and (iv) a privileged network $P$ to support the learning of $T$ by predicting the simulator's PI $z_s$.

recent methods consider the final recognition task during the image translation process. Closely related to our work, PixelDA (Bousmalis et al., 2017) is a pixel-level domain adaptation method that jointly trains a task classifier along with a GAN using simulation as its source domain but no privileged information. These approaches focus on simple tasks and visual conditions that are easy to simulate, hence having a low domain gap to begin with.

On the other hand, Hoffman et al. (2016b) are the first to study semantic segmentation as the task network in adversarial training. Zhang et al. (2017) uses a curriculum learning style approach to reduce domain gap. Saito et al. (2017) conducts domain adaptation by utilizing the task-specific decision boundaries with classifiers. Sankaranarayanan et al. (2018) leverage the GAN framework by learning general representation shared between the generator and segmentation networks. Chen et al. (2018) use a target guided distillation to encourage the task network to imitate a pretrained model. Zhang et al. (2018) propose to combine appearance and representation adaptation. Tsai et al. (2018) propose an adversarial learning method to adapt in the output (segmentation) space. Zou et al. (2018) generates pseudo-labels based on confidence scores with balanced class distribution and propose an iterative self-training framework.

Our main novelty is the use of Privileged Information from a simulator in a generic way by considering a privileged network in our architecture (see Figure 2). We show that for the challenging task of semantic segmentation of urban scenes, our approach significantly improves by augmenting the learning objective with our auxiliary privileged task, especially in the presence of a large sim-to-real domain gap, the main problem in challenging real-world conditions.

Our work is inspired by Learning Using Privileged Information (LUPI) (Vapnik & Vashist, 2009), which is linked to distillation (Hinton et al., 2015) as shown by Lopez-Paz et al. (2015). LUPI's goal is to leverage additional data only available at training time. For unsupervised domain adaptation from a simulator, there is a lot of potentially useful information about the generation process that could inform the adaptation. However, that information is only available at training time, as we do not have access to the internals of the real-world data generator. Several works have used privileged information at training time for domain adaptation (Chen et al., 2014; Hoffman et al., 2016a; Li et al., 2014; Sarafianos et al., 2017; Garcia et al., 2018). Hoffman et al. (2016a) leverage RGBD information to help adapt an object detector at the feature level, while Garcia et al. (2018) propose a similar concept of modality distillation for action recognition. Inspired by this line of work, we exploit the privileged information from simulators for sim-to-real unsupervised domain adaptation.

## 3 SIMULATOR PRIVILEGED INFORMATION GAN

### 3.1 UNSUPERVISED LEARNING WITH A SIMULATOR

Our goal is to design a procedure to learn a model (neural network) that solves a perception task (e.g., semantic segmentation) using raw sensory data coming from a target domain (e.g., videos of

a car driving in urban environments) *without using any ground truth data from the target domain*. We formalize this problem as *unsupervised domain adaptation* from a synthetic domain (source domain) to a real domain (target domain). The source domain consists of labeled synthetic images together with Privileged Information (PI), obtained from the internal data structures of a simulator. The target domain consists of unlabeled images.

The simulated source domain serves as an idealized representation of the world, offering full control of the environment (weather conditions, types of scene, sensor configurations, etc.) with automatic generation of raw sensory data and labels for the task of interest. The main challenge we address in this work is how to overcome the gap between this synthetic source domain and the target domain to ensure generalization of the task network in the real-world *without target supervision*.

Our main hypothesis is that the PI provided by the simulator is a rich source of information to guide and constrain the training of the target task network. The PI can be defined as any information internal to the simulator, such as depth, optical flow, or physical properties about scene components used during simulation (e.g., materials, forces, etc.). We leverage the simulator's PI within a GAN framework, called SPIGAN. Our approach is described in the next section.

## 3.2 SPIGAN

Let $X_r = \{x_r^{(j)}, \ j = 1 \ldots N^r\}$ be a set of $N^r$ unlabeled real-world images $x_r$. Let $X_s = \{(x_s^{(i)}, y_s^{(i)}, z_s^{(i)}), \ i = 1 \ldots N^s\}$ be a set of $N^s$ simulated images $x_s$ with their labels $y_s$ and PI $z_s$. We describe our approach assuming a unified treatment of the PI, but our method trivially extends to multiple separate types of PI.

SPIGAN (cf. Fig. 2) jointly learns a model $(\theta_G, \theta_D, \theta_T, \theta_P)$, consisting of: (i) a generator $G(x; \theta_G)$, (ii) a discriminator $D(x; \theta_D)$, (iii) a task predictor $T(x; \theta_T)$, and (iv) a privileged network $P(x; \theta_P)$. The generator $G$ is a mapping function, transforming an image $x_s$ in $X_s$ (source domain) to $x_f$ in $X_f$ (adapted or fake domain). SPIGAN aims to make the adapted domain statistically close to the target domain to maximize the accuracy of the task predictor $T(x; \theta_T)$ during testing. The discriminator $D$ is expected to tell the difference between $x_f$ and $x_r$, playing an adversarial game with the generator until a termination criteria is met (refer to section 4.1) . The target task network $T$ is learned on the synthetic $x_s$ and adapted $G(x_s; \theta_G)$ images to predict the synthetic label $y_s$, assuming the generator presents a reasonable degree of label (content) preservation. This assumption is met for the regime of our experiments. Similarly, the privileged network $P$ is trained on the same input but to predict the PI $z$, which in turn assumes the generator $G$ is also PI-preserving. During testing only $T(x; \theta_T)$ is needed to do inference for the selected perception task.

The main learning goal is to train a model $\theta_T$ that can correctly perform a perception task $T$ in the target real-world domain. All models are trained jointly in order to exploit all available information to constrain the solution space. In this way, the PI provided by the privileged network $P$ is used to constrain the learning of $T$ and to encourage the generator to model the target domain while being label- and PI-preserving. Our joint learning objective is described in the following section.

## 3.3 LEARNING OBJECTIVE

We design a consistent set of loss functions and domain-specific constraints related to the main prediction task $T$. We optimize the following minimax objective:

$$\min_{\theta_G, \theta_T, \theta_P} \max_{\theta_D} \ \alpha \mathcal{L}_{\text{GAN}} + \beta \mathcal{L}_T + \gamma \mathcal{L}_P + \delta \mathcal{L}_{\text{perc}} \tag{1}$$

where $\alpha$, $\beta$, $\gamma$, $\delta$ are the weights for adversarial loss, task prediction loss, PI regularization, and perceptual regularization respectively, further described below.

**Adversarial loss $\mathcal{L}_{\text{GAN}}$.** Instead of using a standard adversarial loss, we use a least-squares based adversarial loss Mao et al. (2016); Zhu et al. (2017), which stabilizes the training process and generates better image results in our experiments:

$$\mathcal{L}_{\text{GAN}}(D, G) = \mathbb{E}_{x_r \sim \mathcal{P}_r}[(D(x_r; \theta_D) - 1)^2]$$
$$+ \mathbb{E}_{x_s \sim \mathcal{P}_s}[D(G(x_s; \theta_G); \theta_D)^2] \tag{2}$$

where $\mathcal{P}_r$ (resp. $\mathcal{P}_s$) denotes the real-world (resp. synthetic) data distribution.

**Task prediction loss $\mathcal{L}_T$.** We learn the task network by optimizing its loss over both synthetic images $x_s$ and their adapted version $G(x_s, \theta_G)$. This assumes the generator is label-preserving, i.e., that $y_s$ can be used as a label for both images. Thanks to our joint objective, this assumption is directly encouraged during the learning of the generator through the joint estimation of $\theta_P$, which relates to scene properties captured by the PI. Naturally, different tasks require different loss functions. In our experiments, we consider the task of semantic segmentation and use the standard cross-entropy loss (Eq. 4) over images of size $W \times H$ and a probability distribution over $C$ semantic categories. The total combined loss in the special case of semantic segmentation is therefore:

$$\mathcal{L}_T(T, G) = \mathcal{L}_{\text{CE}}(x_s, y_s) + \mathcal{L}_{\text{CE}}(G(x_s; \theta_G), y_s) \tag{3}$$

$$\mathcal{L}_{\text{CE}}(x, y) = \frac{-1}{WH} \sum_{u,v}^{W,H} \sum_{c=1}^{C} \mathbb{1}_{[c=y_{u,v}]} \log(T(x; \theta_T)_{u,v}) \tag{4}$$

where $\mathbb{1}_{[a=b]}$ is the indicator function.

**PI regularization $\mathcal{L}_P$.** Similarly, the auxiliary task of predicting PI also requires different losses depending on the type of PI. In our experiments, we use depth from the z-buffer and an $\ell_1$-norm:

$$\mathcal{L}_P(P, G) = ||P(x_s; \theta_P) - z_s||_1$$
$$+ ||P(G(x_s; \theta_G); \theta_P) - z_s||_1 \tag{5}$$

**Perceptual regularization $\mathcal{L}_{\text{perc}}$.** To maintain the semantics of the source images in the generated images, we additionally use the perceptual loss Johnson et al. (2016); Chen & Koltun (2017):

$$\mathcal{L}_{\text{perc}}(G) = ||\phi(x_s) - \phi(G(x_s; \theta_G))||_1 \tag{6}$$

where $\phi$ is a mapping from image space to a pre-determined feature space Chen & Koltun (2017) (see 4.1 for more details).

**Optimization.** In practice, we follow the standard adversarial training strategy to optimize our joint learning objective (Eq. 1). We alternate between updates to the parameters of the discriminator $\theta_D$, keeping all other parameters fixed, then fix $\theta_D$ and optimize the parameters of the generator $\theta_G$, the privileged network $\theta_P$, and most importantly the task network $\theta_T$. We discuss the details of our implementation, including hyper-parameters, in section 4.1.

## 4 EXPERIMENTS

We evaluate our unsupervised domain adaptation method on the task of semantic segmentation in a challenging real-world domain for which training labels are not available.

As our source synthetic domain, we select the public SYNTHIA dataset (Ros et al., 2016) as synthetic source domain given the availability of automatic annotations and PI. SYNTHIA is a dataset generated from an autonomous driving simulator of urban scenes. These images were generated under different weathers and illumination conditions to maximize visual variability. Pixel-wise segmentation and depth labels are provided for each image. In our experiment, we use the sequence of SYNTHIA-RAND-CITYSCAPES, which contains semantic segmentation labels that are more compatible with Cityscapes.

For target real-world domains, we use the Cityscapes (Cordts et al., 2016) and Mapillary Vistas (Neuhold et al., 2017) datasets. Cityscapes is one of most widely used real-world urban scene image segmentation datasets with images collected around urban streets in Europe. For this dataset, We use the standard split for training and validation with $2,975$ and $500$ images respectively. Mapillary Vistas is a larger dataset with a wider variety of scenes, cameras, locations, weathers, and illumination conditions. We use $16,000$ images for training and $2,000$ images for evaluation. During training, none of the labels from the real-world domains are used.

In our experiment, we first evaluate adaptation from SYNTHIA to Cityscapes on 16 classes, following the standard evaluation protocol used in Hoffman et al. (2016b); Zhang et al. (2017); Saito et al. (2017); Sankaranarayanan et al. (2018); Zou et al. (2018). Then we show the positive impact

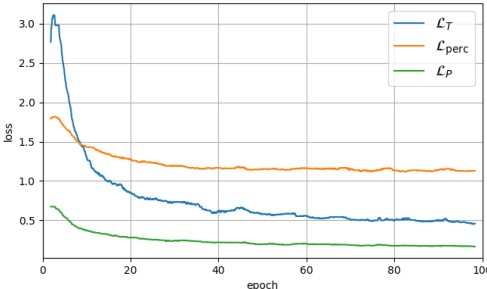
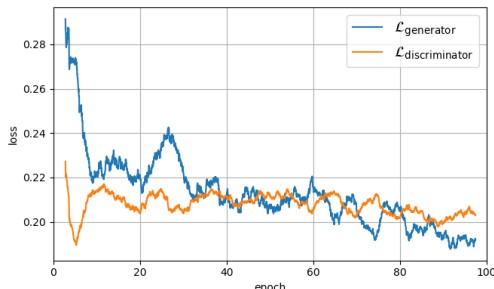

Figure 3: Loss curves for the task, perceptual, and privileged parts of the learning objective during the training of SYNTHIA-to-Cityscapes.

Figure 4: Early stopping at the iteration when the discriminator loss is significantly and consistently better than the generator loss (90 here).

of using PI by conducting ablation study with and without PI (depth) during adaptation from SYNTHIA to both Cityscapes and Vistas, on a common 7 categories ontology. To be consistent with the semantic segmentation best practices, we use standard intersection-over-union (IoU) per category and mean intersection-over-union (mIoU) as our main validation metric.

## 4.1 IMPLEMENTATION DETAILS

We adapt the generator and discriminator model architectures from CycleGAN (Zhu et al., 2017) and (Johnson et al., 2016). For simplicity, we use a single sim-to-real generator (no cycle consistency) consisting of two down-sampling convolution layers, nine ResNet blocks (He et al., 2016) and two fractionally-strided convolution layers. Our discriminator is a PatchGAN (Isola et al., 2017) network with 3 layers. We use the standard FCN8s architecture Long et al. (2015) for both the task predictor $T$ and the privileged network $P$, given its ease of training and its acceptance in domain adaptation works Hoffman et al. (2016b). For the perceptual loss $\mathcal{L}_{\text{perc}}$, we follow the implementation in Chen & Koltun (2017). The feature is constructed by the concatenation of the activations of a pre-trained VGG19 network Witten et al. (2016) of layers "conv1_2", "conv2_2", "conv3_2", "conv4_2", "conv5_2".

Following the common protocol in unsupervised domain adaptation Shrivastava et al. (2016); Zhu et al. (2017); Bousmalis et al. (2016); Sankaranarayanan et al. (2018), we set hyper-parameters using a coarse grid search on a small validation set different than the target set. For Cityscapes, we use a subset of the validation set of Vistas, and vice-versa. We found a set of values that are effective across datasets and experiments, which show they have a certain degree of robustness and generalization. The weights in our joint adversarial loss (Eq. 1) are set to $\alpha = 1$, $\beta = 0.5$, $\gamma = 0.1$, $\delta = 0.33$, for the GAN, task, privileged, and perceptual objectives respectively. This confirms that the two most important factors in the objective are the GAN and task losses ($\alpha = 1$, $\beta = 0.5$). This is intuitive, as the goal is to improve the generalization performance of the task network (the task loss being an empirical proxy) across a potentially large domain gap (addressed first and foremost by the GAN loss). The regularization terms are secondary in the objective, stabilizing the training (perceptual loss) and constraining the adaptation process (privileged loss). Figures 3 and 4 show an example of our loss curves and the stability of our training.

Another critical hyper-parameter for unsupervised learning is the stopping criterion. We observed that the stabilizing effects of the task and privileged losses (Eqs. 3,5) on the GAN objective (Eq. 2) made a simple rule effective for early stopping. We stop training at the iteration when the discriminator loss is significantly and consistently better than the generator loss (iteration 90 in Figure 4). This is inspired by the semi-supervised results of Dai et al. (2017), where effective discriminative adaptation of the task network might not always be linked to the best image generator. We evaluate the methods with two resolutions: $320 \times 640$ and $512 \times 1024$, respectively. Images are resized to the evaluated size during training and evaluation. During training, we sample crops of size $320 \times 320$ (resp. $400 \times 400$) for lower (resp. higher) resolution experiments. In all adversarial learning cases, we do five steps of the generator for every step of the other networks. The Adam optimizer (Kingma & Ba, 2014) is used to adjust all parameters with initial learning rate 0.0002 in our PyTorch implementation (Paszke et al., 2017).

| resolution | Method | road | sidewalk | building | wall | fence | pole | T-light | T-sign | vegetation | sky | person | rider | car | bus | motorcycle | bicycle | mean IoU |
|---|---|---|---|---|---|---|---|---|---|---|---|---|---|---|---|---|---|---|
| 320 × 640 | FCNs wild source-only | 6.4 | 17.7 | 29.7 | 1.2 | 0.0 | 15.1 | 0.0 | 7.2 | 30.3 | 66.8 | 51.1 | 1.5 | 47.3 | 3.9 | 0.1 | 0.0 | 17.4 |
| | FCNs wild | 11.5 | 19.6 | 30.8 | 4.4 | 0.0 | 20.3 | 0.1 | 11.7 | 42.3 | 68.7 | 51.2 | 3.8 | 54.0 | 3.2 | 0.2 | 0.6 | 20.2 |
| | CDA source-only | 5.6 | 11.2 | 59.6 | 0.8 | **0.5** | 21.5 | 8.0 | 5.3 | 72.4 | 75.6 | 35.1 | 9.0 | 23.6 | 4.5 | 0.5 | **18.0** | 22.0 |
| | CDA | 65.2 | 26.1 | **74.9** | 0.1 | **0.5** | 10.7 | 3.7 | 3.0 | 76.1 | 70.6 | 47.1 | 8.2 | 43.2 | 20.7 | 0.7 | 13.1 | 29.0 |
| | LSD source-only | n/a | n/a | n/a | n/a | n/a | n/a | n/a | n/a | n/a | n/a | n/a | n/a | n/a | n/a | n/a | n/a | 23.2 |
| | LSD | n/a | n/a | n/a | n/a | n/a | n/a | n/a | n/a | n/a | n/a | n/a | n/a | n/a | n/a | n/a | n/a | 34.5 |
| | Our source-only | 14.2 | 12.4 | 73.3 | 0.5 | 0.1 | **23.1** | 5.8 | **12.2** | **79.4** | **78.6** | 45.1 | 7.8 | 32.8 | 5.7 | 5.6 | 1.6 | 24.9 |
| | SPIGAN-no-PI | 68.8 | 24.5 | 73.4 | 3.6 | 0.1 | 22.0 | 5.8 | 8.9 | 74.6 | 77.3 | 41.5 | 8.8 | 58.2 | 15.4 | 6.7 | 8.4 | 31.1 |
| | SPIGAN | **80.5** | **38.9** | 73.4 | 1.8 | 0.3 | 20.3 | **7.9** | 11.3 | 77.1 | 77.6 | 46.6 | **13.2** | **63.8** | **22.8** | **8.8** | 11.2 | **34.7** |
| 512 × 1024 | LSD source-only | 30.1 | 17.5 | 70.2 | 5.9 | 0.1 | 16.7 | 9.1 | 12.6 | 74.5 | 76.3 | 43.9 | 13.2 | 35.7 | 14.3 | 3.7 | 5.6 | 26.8 |
| | LSD | 80.1 | 29.1 | **77.5** | 2.8 | **0.4** | 26.8 | 11.1 | **18.0** | 78.1 | 76.7 | 48.2 | **15.2** | 70.5 | 17.4 | 8.7 | 16.7 | 36.1 |
| | CBST source-only | 17.2 | 19.7 | 73.3 | 1.1 | 0.0 | 19.1 | 3.0 | 9.1 | 71.8 | 78.3 | 37.6 | 4.7 | 42.2 | 9.0 | 0.1 | 0.9 | 22.6 |
| | CBST | 69.9 | 28.7 | 69.5 | **12.1** | 0.1 | 25.4 | **11.9** | 13.6 | **82.0** | **81.9** | 49.1 | 14.5 | 66.0 | 6.6 | 3.7 | **32.4** | 35.4 |
| | Our source-only | 21.2 | 12.3 | 69.1 | 2.8 | 0.1 | 24.8 | 10.4 | 15.3 | 74.8 | 78.2 | 50.3 | 8.8 | 41.9 | 18.3 | 6.6 | 6.8 | 27.6 |
| | SPIGAN-no-PI | 69.5 | 29.4 | 68.7 | 4.4 | 0.3 | 32.4 | 5.8 | 15.0 | 81.0 | 78.7 | 52.2 | 13.1 | 72.8 | 23.6 | 7.9 | 18.7 | 35.8 |
| | SPIGAN | 71.1 | **29.8** | 71.4 | 3.7 | 0.3 | **33.2** | 6.4 | 15.6 | 81.2 | 78.9 | **52.7** | 13.1 | **75.9** | **25.5** | **10.0** | 20.5 | **36.8** |
| | (LSD) Target-only | 96.5 | 74.6 | 86.1 | 37.1 | 33.2 | 30.2 | 39.7 | 51.6 | 87.3 | 90.4 | 60.1 | 31.7 | 88.4 | 52.5 | 33.6 | 59.1 | 59.5 |

Table 1: Semantic segmentation unsupervised domain adaptation from SYNTHIA to Cityscapes. We present semantic segmentation results with per-class IoU and mean IoU. The highest IoU (at the same resolution) for each class within the compared algorithms is highlighted with bold font.

| dataset | Method | $\mathcal{L}_{perc}$ | $\mathcal{L}_P$ | flat | const. | object | nature | sky | human | vehicle | mIoU | Neg. Rate |
|---|---|---|---|---|---|---|---|---|---|---|---|---|
| Cityscapes | FCN source | − | − | 79.6 | 51.0 | 8.7 | 29.0 | 50.9 | 3.0 | 31.6 | 36.3 | − |
| | SPIGAN-base | | | 82.5 | 52.7 | 7.2 | 30.6 | 52.2 | 5.6 | 34.2 | 37.9 | 0.34 |
| | SPIGAN-no-PI | ✓ | | 90.3 | 58.2 | 6.8 | 35.8 | 69.0 | 9.5 | 52.1 | 46.0 | 0.16 |
| | SPIGAN | ✓ | ✓ | **91.2** | **66.4** | **9.6** | **56.8** | **71.5** | **17.7** | **60.3** | **53.4** | **0.09** |
| | FCN target | − | − | 95.2 | 78.4 | 10.0 | 80.1 | 82.5 | 37.0 | 75.1 | 65.4 | − |
| Vistas | FCN source | − | − | 61.5 | 40.8 | 10.4 | 53.3 | 65.7 | 16.6 | 30.4 | 39.8 | − |
| | SPIGAN-base | | | 59.4 | 29.7 | 1.0 | 10.4 | 52.2 | 5.9 | 20.3 | 22.7 | 0.83 |
| | SPIGAN-no-PI | ✓ | | 53.0 | 30.8 | 3.6 | 14.6 | 53.0 | 5.8 | 26.9 | 26.8 | 0.80 |
| | SPIGAN | ✓ | ✓ | **74.1** | **47.1** | **6.8** | **43.3** | **83.7** | **11.2** | **42.2** | **44.1** | **0.42** |
| | FCN target | − | − | 90.4 | 76.5 | 32.8 | 82.8 | 94.9 | 40.3 | 77.4 | 70.7 | − |

Table 2: Semantic Segmentation results (per category and mean IoUs, higher is better) for SYN-THIA adapting to Cityscapes and Vistas. The last column is the ratio of images in the validation set for which we observe negative transfer (lower is better).

## 4.2 RESULTS AND DISCUSSION

In this section we present our evaluation of the SPIGAN algorithm in the context of adapting a semantic segmentation network from SYNTHIA to Cityscapes. Depth maps from SYNTHIA are used as PI in the proposed algorithm.

We compare our results to several state-of-art domain adaptation algorithms, including FCNs in the wild (FCNs wild) (Hoffman et al., 2016b), Curriculum DA (CDA) (Zhang et al., 2017), Learning from synthetic data (LSD) (Sankaranarayanan et al., 2018), and Class-balanced Self-Training (CBST) Zou et al. (2018).

Quantitative results for these methods are shown in Table 1 for the semantic segmentation task on the target domain of Cityscapes (validation set). As reference baselines, we include results training only on source images and non-adapted labels. We also provide our algorithm performance without the PI for comparison (i.e., $\gamma = 0$ in Eq. 1, named "SPIGAN-no-PI").

Results show that on Cityscapes SPIGAN achieves state-of-the-art semantic segmentation adaptation in terms of mean IoU. A finer analysis of the results attending to individual classes suggests that the use of PI helps to estimate layout-related classes such as road and sidewalk and object-related classes such as person, rider, car, bus and motorcycle. SPIGAN achieves an improvement of 3% in $320 \times 640$, 1.0% in $512 \times 1024$, in mean IoU with respect to the non-PI method. This improvement is thanks to the regularization provided by $P(x; \theta_P)$ during training, which decreases the number of artifacts as shown in Figure 5. This comparison, therefore, confirms our main contribution: a general approach to leveraging synthetic data and PI from the simulator to improve generalization performance across the sim-to-real domain gap.

### 4.3 ABLATION STUDY

To better understand the proposed algorithm, and the impact of PI, we conduct further experiments comparing SPIGAN (with PI), SPIGAN-no-PI (without PI), and SPIGAN-base (without both PI and perceptual regularization), the task network of SPIGAN trained only on the source domain (FCN source, lower bound, no adaptation), and on the target domain (FCN target, upper bound), all at $320 \times 640$ resolution. We also include results on the Vistas dataset, which presents a more challenging adaptation problem due to the higher diversity of its images. For these experiments, we use a 7 semantic classes ontology to produce a balanced ontology common to the three datasets (SYNTHIA, Cityscapes and Vistas). Adaptation results for both target domains are given in Table 2.

In addition to the conventional segmentation performance metrics, we also carried out a study to measure the amount of negative transfer, summarized in Table 2. A negative transfer case is defined as a real-world testing sample that has a mIoU lower than the FCN source prediction (no adaptation).

As shown in Table 2, SPIGAN-no-PI, including perceptual regularization, performs better than SPIGAN-base in both datasets. The performance is generally improved in all categories, which implies that perceptual regularization effectively stabilizes the adaptation during training.

For Cityscapes, the quantitative results in Table 2 show that SPIGAN is able to provide dramatic adaptation as hypothesized. SPIGAN improves the mean IoU by $17.1\%$, with the PI itself providing an improvement of $7.4\%$. This is consistent with our observation in the previous experiment (Table 1). We also notice that SPIGAN gets significant improvements on "nature", "construction", and "vehicle" categories. In addition, SPIGAN is able to improve the IoU by $+15\%$ on the "human" category, a difficult class in semantic segmentation. We provide examples of qualitative results for the adaptation from SYNTHIA to Cityscapes in Figure 5 and Figure 7.

On the Vistas dataset, SPIGAN is able to decrease the domain gap by $+4.3\%$ mean IoU. In this case, using PI is crucial to improve generalization performance. SPIGAN-no-PI indeed suffers from negative transfer, with its adapted network performing $-13\%$ worse than the FCN source without adaptation. Table 2 shows that $80\%$ of the evaluation images have a lower individual IoU after adaptation in the SPIGAN-no-PI case (vs. $42\%$ in the SPIGAN case).

The main difference between the Cityscapes and Vistas results is due to the difference in visual diversity between the datasets. Cityscapes is indeed a more visually uniform benchmark than Vistas: it was recorded in a few German cities in nice weather, whereas Vistas contains crowdsourced data from all over the world with varying cameras, environments, and weathers. This makes Cityscapes more amenable to image translation methods (including SPIGAN-no-PI), as can be seen in Figure 5 where a lot of the visual adaptation happens at the color and texture levels, whereas Figure 6 shows that SYNTHIA images adapted towards Vistas contain a lot more artifacts. Furthermore, a larger domain gap is known to increase the risk of negative transfer (cf. Csurka (2017)). This is indeed what we quantitatively measured in Table 2 and qualitatively confirmed in Figure 6.

SPIGAN suffers from similar but less severe artifacts. As shown in Figure 6, they are more consistent with the depth of the scene, which helps addressing the domain gap and avoids the catastrophic failures visible in the SPIGAN-no-PI case. This consistent improvement brought by PI in both of the experiments not only shows that PI imposes useful constraints that promote better task-oriented training, but also implies that PI more robustly guides the training to reduce domain shift.

By comparing the results on the two different datasets, we also found that all the unsupervised adaptation methods share some similarity in the performance of certain categories. For instance, the "vehicle" category has seen the largest improvement for both Cityscapes and Vistas. This trend is consistent with the well-known fact that "object" categories are easier to adapt than "stuff" Vazquez et al. (2014). However, the same improvement did not appear in the "human" category mainly because the SYNTHIA subset we used in our experiments contains very few humans. This phenomenon has been recently studied in Sadat Saleh et al. (2018).

## 5 CONCLUSION

We present SPIGAN, a novel method for leveraging synthetic data and Privileged Information (PI) available in simulated environments to perform unsupervised domain adaptation of deep networks. Our approach jointly learns a generative pixel-level adaptation network together with a target task network and privileged information models. We showed that our approach is able to address large

domain gaps between synthetic data and target real-world domains, including for challenging real-world tasks like semantic segmentation of urban scenes. For future work, we plan to investigate SPIGAN applied to additional tasks, with different types of PI that can be obtained from simulation.

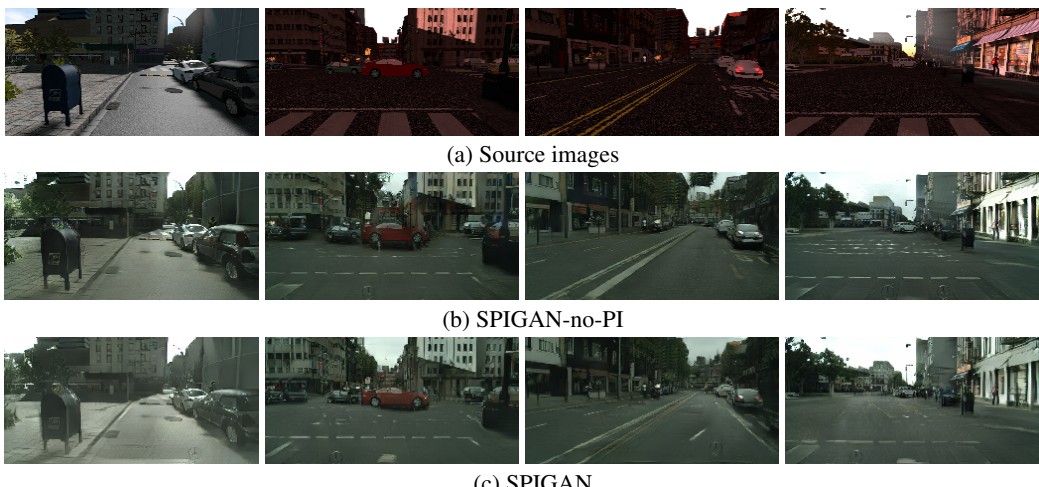

Figure 5: Adaptation from SYNTHIA to Cityscapes. (a) Examples of images from the source domain. (b) Source images after the adaptation process w/o Privileged Information. (c) Source images after the adaptation process using SPIGAN.

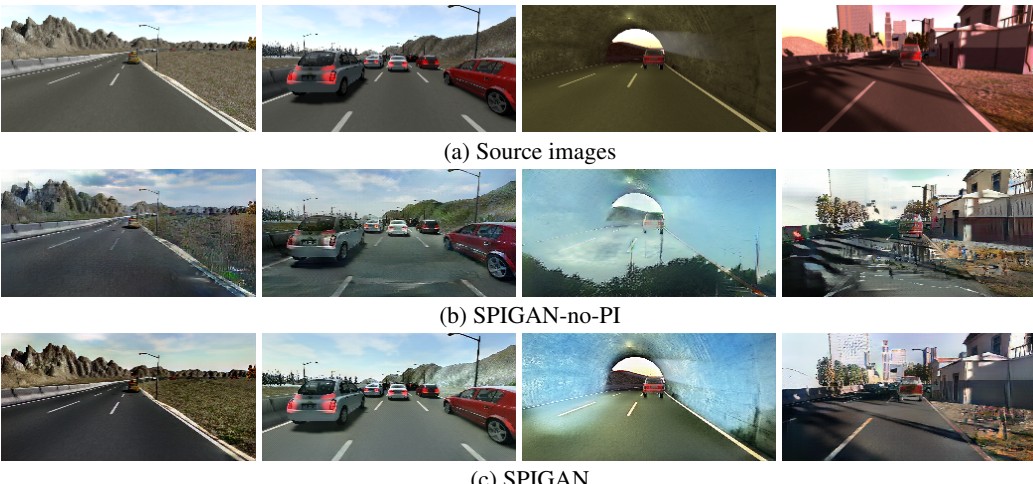

Figure 6: Adaptation from SYNTHIA to Vistas. (a) Examples of images from the source domain. (b) Source images after the adaptation process w/o Privileged Information. (c) Source images after the adaptation process using SPIGAN. Image adaptation is more challenging between these two datasets due to a larger domain gap. Qualitative results indicate that SPIGAN is encoding more regularization in the image generation.

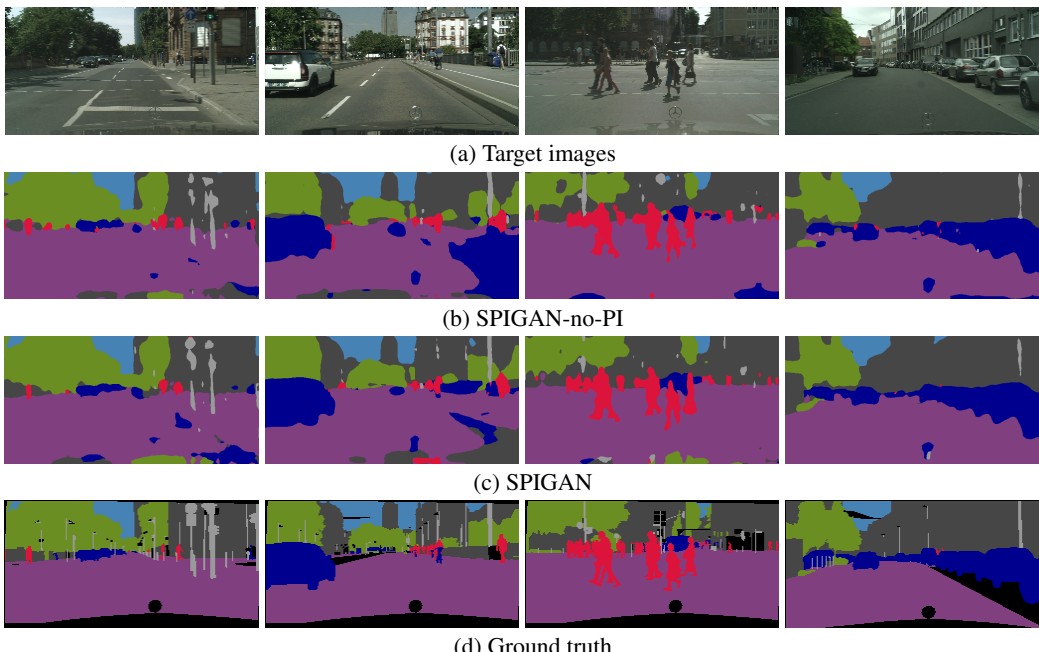

Figure 7: Semantic segmentation results on Cityscapes. For a set of real images (a) we show examples of predicted semantic segmentation masks. SPIGAN predictions (c) are more accurate (i.e., closer to the ground truth (d)) than those produced without PI during training (b).

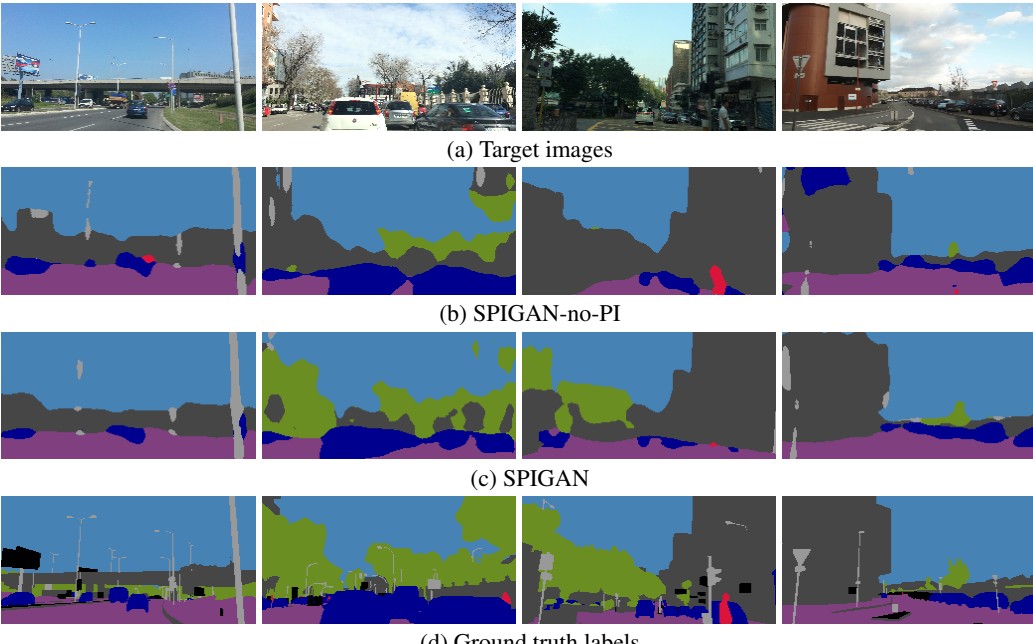

Figure 8: Semantic segmentation results on Vistas. For a set of real images (a) we show examples of predicted semantic segmentation masks. SPIGAN predictions (c) are more accurate (i.e., closer to the ground truth (d)) than those produced without PI during training (b).

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
