# OpenReview forum: "SPIGAN: Privileged Adversarial Learning from Simulation"
_ICLR.cc/2019/Conference_

### Official Review · AnonReviewer1 · 2018-10-29
**Privileged information for domain adaptation**

**Rating:** 7
**Confidence:** 4

**Review:**

This article addresses the problem of domain adaptation of semantic segmentation methods from autonomous vehicle simulators to the real world. The key contribution of this paper is the use of privileged information for performing the adaptation. The method is of those called unsupervised domain adaptation as no labels from the target domain are used for the adaptation. The method is based on a GAN with: a) A generator that transforms the simulation images to real appearance; b) A discriminator that distinguish between real and fake images;  c) a privileged network that learns to perform depth estimation; and d) the task networks that learns to perform semantic segmentation. Privileged information is very few exploited in simulations and I consider it an important way of further exploit these simulators.

The article is clear, short, well written and very easy to understand. The method is effective as it is able to perform domain adaptation and improve over the compared methods. There is also an ablation study to evaluate the contribution of each module. This ablation study shows that the privileged information used helps to better perform the adaptation. The state of the art is comprehensive and the formulation seams correct.  The datasets used for the experiments (Synthia, Cityscapes and Vistas) is very adequate as they are the standard ones.

Some minor concerns:
 - The use of 360x640 as resolution
 - The use of FCN8 instead of something based on Resnet or densenet

I would like some more details on what is happening with Vistas dataset. SPIGAN-no-PI underperforms the source model. By looking at Figure 4 we can observe that the transformation of the images is not working properly as many artifacts appear. In SPIGAN those artifacts does not appear and then the adaptation works better. Could it be a problem in the training?

---

> ### Author Response · Authors · 2018-11-13
> **Authors' response to Reviewer 1**
>
> Thank you very much for your feedback and valuable comments. We are happy you found our submission to be a valuable contribution to the community. We have revised the paper by following your comments, as explained below. All the changes are visible using the "Show Revisions" tool on OpenReview.
>
> Q1: The use of 360x640 as resolution
> This resolution was used for two main reasons: 1) fair comparison (this resolution is part of the standard protocol used by the related works we compare to), 2) faster exploration. Nonetheless, we agree that higher resolution experiments would be interesting. Therefore, following your comments, we ran additional experiments using a much higher resolution (512 x 1024), and got results competitive with the state-of-the-art, reinforcing our previous experimental conclusions. The details are updated in Table 1 in the latest revised version of our manuscript.
>
>
> Q2: The use of FCN8s
> We agree that using bigger and better backbones than FCN8s is likely to result in significant accuracy improvements. We choose to use FCN8s to have a fair comparison with previous domain adaptation works in the literature (cf. Hoffman et al., 2016, Zhang et al., 2017, Sankaranarayanan et al., 2018, and Zou et al. 2018), where FCN8s is widely used. Moreover, we were seeking to simplify our pipeline by reducing the size of the different models that are part of SPIGAN and the time taken to train these models in order to stay within a constrained computational budget for training. In this regard, FCN8s provides us with a simple architecture of low memory footprint and fast to train, which makes exploration easier and faster, thus enabling our ablative analysis. Furthermore, we believe this improves the reproducibility of the paper.
>
>
> Q3: More details on what is happening with Vistas and SPIGAN-no-PI.
> Following your remarks, we have investigated further the difference between our Cityscapes and Vistas results. We could not find any outstanding problem in the training of our baselines or methods: we use the same code, experimental protocol, and parameter tuning in all cases (discussed in more details in the updated Section 4.1). The only difference between SPIGAN-no-PI and SPIGAN is the addition of the PI-based term (Eq.5) in the learning objective (Eq.1). This added term acts as a regularizer, aiming to constrain the optimization to preserve the PI, which is depth information in this case. Our assumption is that this added term improves generalization performance. We in fact run our SPIGAN-no-PI experiments by just setting the PI-regularization hyperparameter \gamma in Eq.1 to 0 and not running the corresponding P network.
>
> Consequently, we believe the difference between Cityscapes and Vistas is indeed explained by the difference between the datasets themselves. Cityscapes is a more visually uniform benchmark than Vistas: Cityscapes was recorded in a few German cities in nice weather, Vistas contains crowdsourced data from all over the world with varying cameras, environments, and weathers. This makes Cityscapes more amenable to image translation methods (including SPIGAN-no-PI), as can be seen in Figure 5 where a lot of the visual adaptation happens at the color and texture levels. Furthermore, a larger domain gap is known to increase the risk of negative transfer (cf. Csurka, G. (2017). Domain adaptation for visual applications: A comprehensive survey. arXiv preprint arXiv:1702.05374.). This is indeed what we quantitatively measured in Table 2 and qualitatively confirmed in Figure 6. SPIGAN-no-PI suffers more from this issue than SPIGAN, which in our view validates our hypothesis: PI improves generalization performance. Note, however, that SPIGAN still suffers from similar but less severe artifacts in Figure 6. They are just more consistent with the depth of the scene, which helps addressing the domain gap and avoids the catastrophic failures visible in SPIGAN-no-PI.
>
> We have clarified the previous points in the revised submission.

---

### Official Review · AnonReviewer3 · 2018-10-30
**[REVISED REVIEW] Interesting way of using depth data from a simulator as Privileged Information.**

**Rating:** 7
**Confidence:** 5

**Review:**

This papers presents an unsupervised domain adaptation algorithm for semantic segmentation. A generative adversarial network is envisaged to carry out synthetic-to-real image translation. In doing so, depth information extracted from a simulator is used as privileged information (PI) to boost the transfer on the target domain, regularizing the model and ensuring a better generalization.

*Quality*
The paper addresses a relevant problem, which is the adaptation of methods from simulated data to real ones. The authors devise a convincing method which takes advantage of state-of-the art generative adversarial architectures and privileged information.

*Clarity*
The paper is sufficiently well written. In general, the main idea and proposed method are clear and easy to follow. The only problem is that some background concepts (such as privileged information or unsupervised domain adaptation) are given for granted, compromising the readability for someone not familiar with those topics.
On a more technical side, for reproducibility purposes, the following aspects have to be clarified:
1.	Details about the validation set used for grid search. Is the validation set extracted from the target domain? (In principled labels from the target domain should not be used during learning).
2.	Number of iterations before convergence: is the training of the network numerically stable? Are there issues in convergence of some of the sub-modules? Which one is leading the learning?
3.	Comments about the relative magnitudes of losses. This will maybe give some intuitions about the values used for the hyper-parameters (e.g., the L_PI is only weighted by 0.1).

*Originality*
The way authors take advantage of depth information extracted from a simulator as privileged information is novel in the sense that, with respect to the original student-teacher paradigm of the paper by Vapnik & Vashist, here the idea of privileged information is interpreted as a regularizer to boost the training stage.

*Significance*
The application of semantic segmentation in urban scenes for navigation tasks is relevant. The scored results are on pair with/ superior to state-of-the-art in unsupervised domain adaptation.
However, the ablation study could be more extensive in order to understand the contribution of the several components, besides the PI network. In fact, it would be interesting to analyze the contribution of the perceptual loss (and others). Also, one could include the target-only result (as done in original LSD paper) to provide an upper bound on the best accuracy that is achievable.

*Pros*
1. The applicative setting of semantic segmentation in urban scenes for navigation is relevant.
2. Using privileged information from simulators seems novel and well presented in this paper.
3. Strong experimental results achieved in challenging benchmarks.

*Cons*
1. The regularization effect of the PI network could be supported by a more extensive ablation study of the model, for example by ablating the several losses used (in particular, the perceptual loss).
2. A quite relevant amount of hyper-parameters need to be cross-validated. Is the method robust against different parameters’ configuration?
3. Missing citations [1, 2]: there are works in the literature that can hallucinate a missing modality during testing. Although such works approach a different problem, authors should cite them.

[1] Judy Hoffman, Saurabh Gupta, Trevor Darrell - Learning with Side Information through Modality Hallucination – CVPR 2016
[2] Nuno Garcia, Pietro Morerio, Vittorio Murino - Modality Distillation with Multiple Stream Networks for Action Recognition – ECCV 2018

*Final Evaluation*
The authors face the challenging synthetic-to-real adaptation setup, with an interesting usage of z-buffer from a simulator as privileged information. Overall, the work is fine, apart from the following points.
1.	In addition to a few missing citations [1, 2], an ablation study on the perceptual loss is necessary to dissect the impact of each component of the pipeline.
2.	The clarity of the paper can be improved by adding some background material on unsupervised domain adaptation and learning with privileged information (PI), as to better highlight the technical novelty of using PI within a L1 regularizer.
3.	The training stage of all submodules could have better investigated, for instance, by providing some convergence plots of the loss functions across iterations.
4.	How to do grid search for parameters in a domain adaptation setting is always a delicate aspect and authors seem elusive on that respect.
5.	Again about hyper-parameters. Due to their high number, some sensitivity analysis should have provided.
As it is, the paper’s strengths slightly outperform the weaknesses, leading to an overall borderline-accept. If authors implement the suggested modification, a full acceptance will be feasible.

[COMMENTS AFTER AUTHORS' RESPONSE]
After the rebuttal provided by authors, all raised questions and criticisms have been fully solved. Therefore, I recommend for a full acceptance.

---

> ### Author Response · Authors · 2018-11-13
> **Authors' response to Reviewer 3**
>
> Thank you for your very detailed review and generous feedback towards making our submission even stronger. We are happy you found our work on this challenging problem valuable and novel. We have revised the paper by following your comments, as described in more details below. All the changes are visible using the "Show Revisions" tool on OpenReview.
>
> Q1: Missing citations
> Thank you for pointing out the missing citations, which are relevant indeed. We have added and discussed briefly the provided references in the revised version of the related work.
>
> Q2: Ablation study on perceptual loss
> We agree this is an interesting additional experiment to run to have a completely thorough ablative analysis. We did not initially run it, because the focus of the analysis is on measuring the relative importance of our contribution (PI), which is why we discussed only SPIGAN-no-PI vs SPIGAN, both using the perceptual loss. We will add results for SPIGAN-no-PI without perceptual loss in the next revised version (we are currently running these additional experiments).
>
> Q3: Clarity of the paper
> Thank you for the detailed suggestions. We have added related background material in section 2 in the current revised version. We have also added the target-only results to both Table 1 and Table 2.
>
> Q4: Convergence plots and training analysis
> We have added the loss curves and a related discussion in section 4.1, confirming the stability of our training regime.
>
> Q5: validation set + hyper-parameters
> Thank you for pointing out a part of our main text that needs to be clarified. Setting hyper-parameters in a fully unsupervised setting is challenging indeed. As you mention, we ensured we do not use any labels from the target dataset. We follow the common protocol in unsupervised domain adaptation [Shrivastava et al., 2016, Zhu et al., 2017, Bousmalis et al., 2017, Sankaranarayanan et al., 2018]: we tune hyper-parameters using grid search on a small validation set different than the target set. For Cityscapes, we use a subset of the validation set of Vistas, and vice-versa. Note that the values found are the same across datasets and experiments, which shows they have a certain degree of robustness and generalization. We have added a clarification in section 4.1.
>
> Moreover, our hyper-parameters described in section 4.1 confirmed that the two most important factors in the objective are the GAN and task losses (\alpha=1, \beta=0.5). This is intuitive, as the goal is to improve the generalization performance of the task network (the task loss being an empirical proxy) across a potentially large domain gap (addressed first and foremost by the GAN loss). At a secondary level of importance come the regularization terms of the loss, which in our case are “contents-preserving” related: 1) the perceptual loss, which accounts for the semantics of the scene (is used for stabilizing the GAN training as mentioned in Shrivastava et al.), and 2) our PI loss, which accounts for the geometry of the scene and is an additional constraint on the adaptation. This is again intuitive, as the regularizers are not the main learning objective. The right balance of these two type of “content”-preserving factors was found via grid search as described above. We have added the details and loss curves in section 4.1 in the revised version.

---

> > ### Author Response · Authors · 2018-11-23
> > **A revised version with additional experiments (ablation study) is available.**
> >
> > We have added the experimental results for SPIGAN-no-PI without perceptual loss, named SPIGAN-base in our latest revised version. As shown in Table 2, we observe that SPIGAN-no-PI (with perceptual loss) outperforms SPIGAN-base (without perceptual loss) in both datasets. This implies that perceptual regularization indeed helps stabilizing the adaptation during training, as suggested in Shrivastava et al., (2016) and Bousmalis et al., (2017). Furthermore, SPIGAN-base only improves over the source model by +1.6% mIoU on Cityscapes and has slightly worse negative transfer than SPIGAN-no-PI on Vistas, whereas the full SPIGAN significantly improves performance as previously discussed. This provides further evidence to back our claim: regularization, including our PI-based term, is indeed a key component to improve generalization performance in domain adaptation.

---

### Official Review · AnonReviewer2 · 2018-11-09
**interesting use of depth information from simulators as priviledged information for unsupervised domain adaptive segmentation**

**Rating:** 6
**Confidence:** 5

**Review:**

The paper focuses on the problem of semantic segmentation across domains. The most standard setting for this task involves real world street images as target and synthetic domains as sources with images produced by simulators of photo-realistic hurban scenes.  This work proposes to leverage further depth information which is actually produced by the simulator together with the source images but which is in general not taken into consideration.
The used deep architecture is a GAN where the generator learning is guided by three components: (1) the standard discriminator loss (2) the cross entropy loss for image segmentation that evaluates the correct label assignment to each image pixel (3) an  l1-based loss which evaluates the correct prediction of the depth values in the original and generated image. A further perceptual regularizer is introduced to support the learning.

+ overall the paper is well organized and easy to read
+ the proposed idea is smart: when starting from a synthetic domain there may be several hidden extra information that are generally neglected but that can instead support the learning task
+ the experimental results seem promising

Still, I have some concerns

- if the main advantage of the proposed approach is in the introduction of the priviledged information, I would expect that disactivating the related PI loss we should get back to results analogous of those obtained by other competing methods. However from Table 2 it seems that SPIGAN-no-PI is already much better than the  FCN Source baseline in the Cityscape case and much worse in the Vistas case. This should be better clarified -- are the basic structure of SPIGAN and FCN analogous?

- the ablation does not cover an analysis on the role of the perceptual regularizer. This is also related to the point above: the use of a perceptual loss may introduce a basic difference with respect to competing methods. It should be better discussed.

- section 4.1 mentions the use of a validation set. More details should be provided about it and on how the hyperparameters were chosen.
A possible analysis on the robustness of the method to those parameters could provide some further intuition about the network stability.
It might be also interesting to check if the  the loss weights provide some intuition  about the relative importance of the losses in the learning process.

- the negative transfer rate is another way to measure the advantage of using the PI with respect to not using it. However, since it is not evaluated for the competing methods its value does not add much information and indeed it is only quickly mentioned in the text. It should be better discussed.

- some recent papers have shown better results than those considered here as baseline:
[Learning to Adapt Structured Output Space for Semantic Segmentation, CVPR 2018]
[Unsupervised Domain Adaptation for Semantic Segmentation via Class-Balanced Self-Training, ECCV 2018]
they should be included as related work and considered as reference for the experimental results.

Overall I think that the proposed idea is valuable but the paper should better clarify the points mentioned above.

---

> ### Author Response · Authors · 2018-11-13
> **Authors' response to Reviewer 2**
>
> Thank you very much for your precise comments and suggestions. We are delighted you found our idea smart, valuable, and our results promising. We have revised the paper by answering your comments as described below.
>
> Q1: SPIGAN-no-PI better than FCN on Cityscapes, worse on Vistas + basic structures of SPIGAN and FCN
> Thank you for your detailed questions. We have clarified the following in the updated submission.
>
> SPIGAN's task network (T in Fig.2) is exactly the FCN network used as baseline in Table 2. At test time, we run only this task network T, which differs only by its weights from the FCN baseline. In the case of the FCN baseline, these weights are trained in a supervised fashion in simulation (source only). In the case of SPIGAN, the task network's weights are obtained via our unsupervised domain adaptation algorithm (using sim and unlabeled target data), with the goal of improving generalization performance over the domain gap. This explains why SPIGAN's results are better than FCN's.
>
> SPIGAN-no-PI also performs domain adaptation, but does not use Privileged Information (PI), which we postulate is helpful. SPIGAN-no-PI improves generalization performance over FCN (no adaptation) on Cityscapes, but not on Vistas. We measured this phenomenon, called negative transfer in the Domain Adaptation literature, in Table 2 (last column) and qualitatively visualized it in Figures 5-8. These results confirm that PI indeed helps, as SPIGAN improves generalization performance overall (better mIoU) and reduces individual negative transfer cases. The root cause of the difference in behavior of SPIGAN-no-PI between Cityscapes and Vistas is discussed in more details in section 4.3, and in the response to Reviewer 1. It is due to a larger visual variety in Vistas than in Cityscapes.
>
> Q2: Ablation study on perceptual loss
> This is an interesting additional experiment to run. We did not initially run it, because the focus of the analysis is on measuring the relative importance of our contribution (PI), which is why we discussed only SPIGAN-no-PI vs SPIGAN, both using the perceptual loss. We will add results for SPIGAN-no-PI without perceptual loss in the next revised version (we are currently running these additional experiments).
>
> Q3: Validation set + hyper-parameters
> Thank you for pointing out a part of our main text that can be clarified. We follow the common protocol in unsupervised domain adaptation [Shrivastava et al., 2016, Zhu et al., 2017, Bousmalis et al., 2017]: we tune hyper-parameters using grid search on a small validation set different than the target set. For Cityscapes, we use a subset of the validation set of Vistas, and vice-versa. Note that the values found are the same across datasets and experiments, which shows they have a certain degree of robustness and generalization. We have added a clarification in section 4.1.
>
> Moreover, our hyper-parameters described in section 4.1 confirmed that the two most important factors in the objective are the GAN and task losses (\alpha=1, \beta=0.5). This is intuitive, as the goal is to improve the generalization performance of the task network (the task loss being an empirical proxy) across a potentially large domain gap (addressed first and foremost by the GAN loss). At a secondary level of importance come the regularization terms in the objective: 1) the perceptual loss (for stabilizing the GAN training), and 2) our PI loss, which is an additional constraint on the adaptation. This is again intuitive, as the regularizers are not the main learning objective. We have added details and loss curves in section 4.1 in the revised version.
>
> Q4: Better discuss negative transfer rate
> Thank you for your suggestion. We believe Table 2 and Figures 5-8 quantitatively and qualitatively describe an important causal explanation for our mean IoU results: the relative importance of instances with negative transfer, an important failure mode of domain adaptation methods in general (cf. Csurka, G. (2017): Domain adaptation for visual applications: A comprehensive survey). Previous related works we compare to do not measure this phenomenon or discuss it in depth, hence why we proposed this new complementary measure and only limited the discussion to our ablative analysis. We expanded on this point in section 4.3, and hope our negative transfer metric will encourage other researchers to discuss negative transfer in more depth and compare to our results.
>
> Q5: Two missing related papers:
> Thank you for pointing out these missing citations. We discuss these two works in section 2 in our revised version. For fair comparison, we only listed the second paper's results in the updated Table 1, because the Synthia-to-Cityscapes results in the first paper are based on a reduced ontology, while all other methods (including ours) report results on 16 classes. Our method outperforms Zhou et al when using the same resolution and FCN8s task network.

---

> > ### Author Response · Authors · 2018-11-23
> > **A revised version with additional experiments (ablation study) is available.**
> >
> > We have added the experimental results for SPIGAN-no-PI without perceptual loss, named SPIGAN-base in our latest revised version. As shown in Table 2, we observe that SPIGAN-no-PI (with perceptual loss) outperforms SPIGAN-base (without perceptual loss) in both datasets. This implies that perceptual regularization indeed helps stabilizing the adaptation during training, as suggested in Shrivastava et al., (2016) and Bousmalis et al., (2017). Furthermore, SPIGAN-base only improves over the source model by +1.6% mIoU on Cityscapes and has slightly worse negative transfer than SPIGAN-no-PI on Vistas, whereas the full SPIGAN significantly improves performance as previously discussed. This provides further evidence to back our claim: regularization, including our PI-based term, is indeed a key component to improve generalization performance in domain adaptation.

---

### Meta-Review · Area_Chair1 · 2018-12-11
**A new approach to learning from simulated data with privileged information**

**Confidence:** 5
**Recommendation:** Accept (Poster)

**Metareview:**

The paper proposes an unsupervised domain adaptation solution applied for semantic segmentation from simulated to real world driving scenes. The main contribution consists of introducing an auxiliary loss based on depth information from the simulator. All reviewers agree that the solution offers a new idea and contribution to the adaptation literature. The ablations provided effectively address the concern that the privileged information does in fact aid in transfer. The additional ablation on the perceptual loss done during rebuttal is also valuable and should be included in the final version.

The work would benefit from application of the method across other sim2real dataset tasks so as to be compared to the recent approaches mentioned by the reviewers, but the current evaluation is sufficient to demonstrate the effectiveness of the approach over baseline solutions.